# Screening for the $^{137}$Cs body burden owing to the Chernobyl accident in Zhytomyr region, Ukraine: 2009–2018

**Yesbol Sartayev[1], Jumpei Takahashi[2], Alexander Gutevich[3], Naomi Hayashida[1,4]***

**1** Department of Radiation and Environment Health Effects, Nagasaki University Graduate School of Biomedical Sciences, Nagasaki, Japan, **2** Center for International Collaborative Research, Nagasaki University, Nagasaki, Japan, **3** Zhytomyr Inter-Area Medical Diagnostic Center, Korosten, Ukraine, **4** Division of Strategic Collaborative Research, Atomic Bomb Disease Institute, Nagasaki University, Nagasaki, Japan

* naomin@nagasaki-u.ac.jp

**Data Availability Statement:** Data cannot be shared publicly because of the ethical restrictions on sharing data publicly. The optout of this study

## Abstract

The Chernobyl Nuclear Power Plant accident released vast amounts of various fission products, of which $^{131}$I and $^{137}$Cs are considered the most important because they are a major source of radiation exposure to the general public. $^{137}$Cs, unlike $^{131}$I, has a 30-year half-life that continues to expose people internally and externally for several decades after being deposited into the ground. $^{137}$Cs can be easily transmitted to the body through the intake of products and wild forest foodstuffs produced in contaminated areas. In the early phase, external exposure to $^{137}$Cs was predominant; however, it gradually diminished, mostly owing to horizontal and vertical distribution, and internal exposure started gaining dominance. Prior studies have shown that people in the affected areas have been constantly subjected to the inevitable low-grade internal exposure, which in turn has led to high anxiety and concern regarding the potential health effects. The present 10-year study assessed the latest status of the body burden among residents of the contaminated parts of the Zhytomyr region of Ukraine by measuring the internal concentration of $^{137}$Cs using the whole-body counter installed at the Medical Center of Korosten city. Almost 110 000 examinees from eight different districts were screened between 2009 and 2018. The study area was situated to the west of Chernobyl, which experienced a significant fallout of $^{137}$Cs and $^{131}$I from the Chernobyl Nuclear Power Plant. We analyzed the data for internal exposure to $^{137}$Cs by obtaining the actual magnitude, pace, and pattern of change for each year. During the study, the average $^{137}$Cs concentration in residents fell from 21.6 Bq/kg at the beginning of the study to 3.0 Bq/kg at the end of the study. The proportion of examinees with detectable levels had also fallen rapidly, from 45% to 11%. We found a weaker seasonal effect and a significantly higher Bq/kg concentration in adolescents than in other age groups.

## Introduction

More than 30 years have elapsed since the accident at the Chernobyl Nuclear Power Plant (CNPP) that created the worst consequences for health and the environment in the history of

does not include a provision for publicly sharing data. Thus, the Ethics Committee of Nagasaki University is restricting these data. Furthermore, because the dataset was provided by the undisclosed database of the Research Center for Radiation Disaster Medical Science (https://home.hiroshima-u.ac.jp/housai/index-e.html. Tel.: +81−95−819−7198) only for conducting this research, an anonymous dataset cannot be made publicly available. Data are available from the Research Center for Radiation Disaster Medical Science (https://home.hiroshima-u.ac.jp/housai/index-e.html) for researchers who meet the criteria for access to the data. Requests for the data set require permission from the research ethics review board at the organization of their affiliation and the Ethics Committee at Nagasaki University Graduate School of Biomedical Sciences. The contact information for the ethics committee is as follows: Tel.: +81−95−819−7198, Email: gakujutu_gakuji@ml.nagasaki-u.ac.jp Website: http://www.mdp.nagasaki-u.ac.jp/research/support_rinri.html.

**Funding:** This research was supported by financial support from the Program of the Network-type Joint Usage/Research Center for Radiation Disaster Medical Science (URL: https://home.hiroshima-u.ac.jp/housai/index-e.html) and the Atomic Bomb Disease Institute of Nagasaki University (URL: https://www-sdc.med.nagasaki-u.ac.jp/abdi/index.html). The funders had no role in study design, data collection and analysis, decision to publish, or preparation of the manuscript.

**Competing interests:** The authors have declared that no competing interests exist.

the nuclear power plant and humankind. More than 30 first responders died owing to the acute high-dose radiation sustained while extinguishing the fire immediately after the explosion. Furthermore, thousands of children from the contaminated zones were diagnosed with thyroid cancer. Various studies have reported that several hundred thousand liquidators have been exposed to high doses of radiation, and millions of people residing in contaminated areas have been experiencing chronic exposure to low radiation doses. This accident released huge amounts of various artificial radionuclides, consisting predominantly of [137]Cs, [134]Cs, and [131]I. These radionuclides caused major contamination of extensive areas of land in Ukraine, Belarus, Russia, and some states of Europe, which in turn negatively impacted the lives of residents living in the contaminated territories [1].

[131]I has an 8-day half-life that almost disappears naturally in a few weeks while [134]Cs and [137]Cs have longer half-lives of 2.1 and 30 years, respectively. Therefore, although [137]Cs continues to irradiate people in the vicinity of the CNPP, externally and internally, it does so at a lower level than that during the initial phase, as it has been partly decayed and redistributed over time. Several years after the accident (1986–1990), the average total effective dose was reduced by approximately 80%, mainly owing to the countermeasures undertaken on a huge scale by the governments. However, future reductions in exposure levels are expected to be rather slow, that is, approximately 3% to 5% per year, though a great majority of the dose from the accident has already been accumulated [2]. Jelin summarized several studies dedicated to investigating contaminated territories; he focused on assessing the concentration of [137]Cs in the body, pointing out that *"[137]Cs is considered to be the greatest ongoing environmental hazard to human health by radiologists and public health scientists"* owing to its long-lived gamma and beta-emitting characteristics [3]. According to the International Atomic Energy Agency (IAEA), a vast majority of the 5 million people residing in the contaminated areas of Belarus, Russia, and Ukraine had received annual effective doses of less than 1 mSv from [137]Cs released after the accident at CNPP in addition to the naturally-occurring background radiation. However, approximately 100 000 residents of the highly contaminated areas still receive more than 1 mSv annually [1].

Several studies devoted to the measurement of the internal body burden of [137]Cs in the late 90s and early 2000s have reported a large proportion of people with detectable internal effective doses, the proportion of which usually varied from one-half to one-third of the examinees from the contaminated areas of Ukraine and Russia [4–9]. Researchers in the field of radiation exposure have emphasized that the average effective dose rate due to internal exposure was becoming increasingly important because it decreases more slowly than that due to external exposure [4, 8]. The dose rate due to internal exposure changes in a complicated and chaotic way, often varying to a large extent without any specific precondition or pattern, which makes it difficult to predict or foresee, unlike that due to external exposure.

Many studies are still ongoing in different affected regions, with the aim of determining the current status, pace of change, and aftermath of the accident, specifically focusing on the possible adverse effects to health internally and externally. Many researchers have reported that internal exposure has been decreasing over the years, though there were significant fluctuations with no accurately identified reasons. These differences can be attributed to the changes in the socioeconomic status of the countries, plentiful years of forest food, heterogeneity of [137]Cs behavior in the contaminated soil, and dietary habits of the affected residents [10].

Our study aimed to assess and identify the overall current status of the internal body burden of residents living in Korosten city and its vicinity and to explain to some extent the relevant changes and findings revealed during the study. We conducted the present study using data collected from the whole-body counter (WBC), which measured the internal [137]Cs concentration of nearly 110 000 people over a period of 10 years. The main objective of the investigation

was to assess the magnitude of internal exposure, its pace, patterns of change, and the factors that may have affected the annual changes in the body burden during the given time period.

## Materials and methods

The research was conducted in the Zhytomyr region, Ukraine, which includes Korosten city and its eight subordinate districts: Olevsk, Emelchinsky, Ovruchsky, Luginsky, Korostenski, Volodar-Volynsky, Narodichsky, and Malinsky. The population in this territory of approximately 323 000 people as of January 1, 2019, generally sought healthcare services at the Medical Center in Korosten city. This area is situated to the west of the CNPP (Korosten city and the eight districts). The closest village included in our study was located 40 km from CNPP, while the most populous Korosten city was 120 km away from the CNPP. This area was among those heavily affected by the fallouts from the CNPP. In Ukraine, four contamination zones were defined depending on the deposition of $^{137}$Cs in the soil; zone I with deposition of $^{137}$Cs above 1480 kBq/m$^2$ had the highest contamination and was named the "exclusion zone," zone II had a deposition of 1480–555 kBq/m$^2$ and named the "obligatory resettlement" zone (although some people continued living within the zone), and zones III and IV had depositions of 555–185 kBq/m$^2$ and 185–33 kBq/m$^2$, respectively, with both permitting living in the areas with the warning that exposures may exceed 1 mSv/y and 0.5 mSv/y in the respective zones. A major part of the settlements in our study belonged to zones III and IV, with only a few villages assigned as level 2 contamination zones.

All participants of the study were residents of Korosten city and the eight subordinate districts, who voluntarily underwent WBC screening at the Zhytomyr Inter-Area Medical Diagnostic Center (Medical Center) to measure their internal $^{137}$Cs concentration free of charge. Measurement results of internal exposure were collected from January 2009 to November 2018, solely at the Medical Center, as it was properly equipped with an appropriate WBC and attended by competent professional medical staff.

The level of internal $^{137}$Cs concentration was measured using a WBC manufactured by Aloka Co., Ltd. (Japan), equipped with a NaI (TI) detector 7.6 cm in diameter and 7.6 cm in thickness, with a 5 cm-thick lead shield. The WBC had a sliding height- and angle-adjustable seat so that the examinee could lean his or her abdomen onto the detector. The seat and back of the chair were shielded by lead plates. The minimum level of $^{137}$Cs that can be detected in the body is 270 Bq per body. Gamma rays emitted from the examinee's body were detected and analyzed using a 240-channel spectrometer.

The results of the measurements were automatically saved on a personal computer (PC) connected to the WBC and simultaneously recorded in a log book to preserve the data. For a period of time, the connection and data recording settings of the PC malfunctioned. During this time, the values of measurements were not automatically saved and the omitted data were retrieved from the log books and added to an Excel file for data analysis. The collected data included the name, sex, date of birth, weight, Bq per body, date of examination, residential address, and other additional information necessary for the calibration of the machine. We only used data necessary for our analysis.

The measurements obtained from the detector required additional adjustments for body weight according to the manufacturer's instructions. Additional calculations were used to estimate the annual effective exposure dose using the effective dose coefficient, $2.5 \times 10^{-3}$ mSv/y per Bq/kg [8]. To assess the seasonal effect, seasons were divided into four seasonal periods: spring (March 1–May 31), summer (June 1–August 31), autumn (September 1–November 30), and winter (December 1–February 28). The data set was divided into four different groups based on age: 0–10, 11–20, 21–60, and > 60 years, with the aim of comparing the mean

concentrations in each age group. The reason for the categorization of the data set into four groups was owing to the different life activities and occupations in each group as well as human growth progress (children, adolescents, adults, and the elderly) to understand the effect in each group.

Data are presented as means and standard deviations (SD), 75$^{th}$ and 95$^{th}$ percentiles of Bq/kg, and proportions of various groups with detectable levels. The participants' $^{137}$Cs exposure was classified as "0 Bq" when it was undetectable. IBM SPSS Statistics software version 25.0 for MacOS (SPSS Japan, Tokyo, Japan) was used for all statistical analyses. The statistical significance of the mean values was determined using the Mann–Whitney U test and analysis of variance followed by the Tamhane test, after non-equal variance using Levene's test was confirmed. To test the significance of frequencies, the chi-square non-parametric test was used, after which a post hoc test was applied when necessary. Pearson's correlation test was used to analyze correlations. P values less than 0.05 indicated statistical significance.

In accordance with the *"Ethical Guidelines for Medical and Health Research Involving Human Subjects"* published by the Ministry of Education, Culture, Sports, Science, and Technology and the Ministry of Health, Labour, and Welfare, this study was approved by the Ethics Committee at Nagasaki University Graduate School of Biomedical Sciences (approval no.: 19042601). Informed consent was obtained in the form of opt-out on the website of Nagasaki University (http://www.mdp.nagasaki-u.ac.jp/research/disclosure2.html).

## Results

Throughout this 10-year study, which was carried out from January 2009 to November 2018, 109,936 residents were screened for internal doses of $^{137}$Cs. The number of residents varied moderately each year, from the smallest group consisting of 9,014 people to the largest consisting of 12,987 people in 2009 and 2012, respectively, as shown in Table 1. The average age for the entire data set was approximately 41 years, with an average approximate weight of 70 kg. The proportion of women for the entire data set was nearly 71% of the population.

The $^{137}$Cs concentrations in the body shown in Fig 1 indicates the value of Bq/kg concentration at the 75$^{th}$ and 95$^{th}$ percentiles of the participants as well as the mean Bq/kg for each study year. A significantly higher annual average Bq/kg concentration level was detected in 2009 (p<0.001) than that in all other years, except 2011 and 2012, when the difference was not significant. It then declined gradually over the course of the study, with one abrupt decline in 2010. The Bq/kg concentration at the 75$^{th}$ percentile was detectable in only the first 3 years of the study, with values of 23.9, 17.4, and 12.0 Bq/kg in 2009, 2011, and 2012, respectively. However, the value at the 95$^{th}$ percentile was detectable in each year of the study and it exhibited

**Table 1.  Population, age, sex, and weight of the whole data set by year.**

|  | **All** | **2009** | **2010** | **2011** | **2012** | **2013** | **2014** | **2015** | **2016** | **2017** | **2018** |
|---|---|---|---|---|---|---|---|---|---|---|---|
| No. | 109,936 | 9,014 | 12,071 | 11,337 | 10,478 | 9,804 | 11,005 | 10,904 | 11,562 | 12,987 | 10,774 |
| **Age** | | | | | | | | | | | |
| Mean | 41 | 39 | 38 | 43 | 43 | 39 | 40 | 41 | 41 | 43 | 44 |
| SD | 18.9 | 19.3 | 18.7 | 18.6 | 18.0 | 19.1 | 18.9 | 19.0 | 19.0 | 19.0 | 18.5 |
| **Women** | | | | | | | | | | | |
| No. | 77,848 | 5,859 | 9,105 | 7,943 | 7,245 | 7,065 | 7,788 | 7,698 | 8,234 | 9,272 | 7,639 |
| (%) | 70.8 | 65.0 | 75.4 | 70.1 | 69.1 | 72.1 | 70.8 | 70.6 | 71.2 | 71.4 | 70.9 |
| **Weight (kg)** | | | | | | | | | | | |
| Mean | 69.7 | 60.2 | 69.9 | 67.8 | 72.0 | 69.9 | 70.2 | 70.4 | 70.9 | 71.6 | 72.0 |
| SD | 19.9 | 19.2 | 19.9 | 19.8 | 19.3 | 20.6 | 20.2 | 19.8 | 19.8 | 19.5 | 18.4 |

nearly the same pattern as the mean Bq/kg for all the years of the study, showing a significant correlation (r = 0.855, p = 0.002).

The frequency of people with detectable levels of $^{137}$Cs decreased sustainably along with the progress of the study, with notable fluctuations in some years (p<0.001). In the first year of study, 44.6% of the residents had detectable levels of $^{137}$Cs, and their frequency gradually declined to 11.2% in 2018, as presented in Fig 2, with one abrupt decline in 2010.

Table 2 shows the number of people and their percentage for a specified range of effective dose (mSv/y) for each year from 2009 to 2018. The percentage of people with an effective dose less than 0.1 mSv/y increased steadily and accounted for 86% and 99% in 2009 and 2018, respectively (p<0.001). The post hoc test showed significance for all years in only two groups, the < 0.1 mSv and ≤ 1 mSv groups. People with a dose > 1 mSv/y declined to a large extent immediately after the first year, with no cases after 2016. Only seven participants were observed with a dose > 5 mSv/y across the entire 10-year study, of which six were aged > 40 years and one was a boy aged < 15 years; they were from various locations in the study domain.

The seasonal changes in $^{137}$Cs concentration in the body are shown in Table 3. The aggregated mean of the body burden in autumn and winter was higher, though there was no evidence that they were significant (p>0.05). Even though the mean Bq/kg stratified by years was slightly higher in autumn and winter, the difference was not significant.

We conducted a frequency analysis of people with detectable levels of $^{137}$Cs in each season for each independent year to identify and gauge the existence of seasonality from a different perspective. The frequency of people with detectable levels in each season was derived from

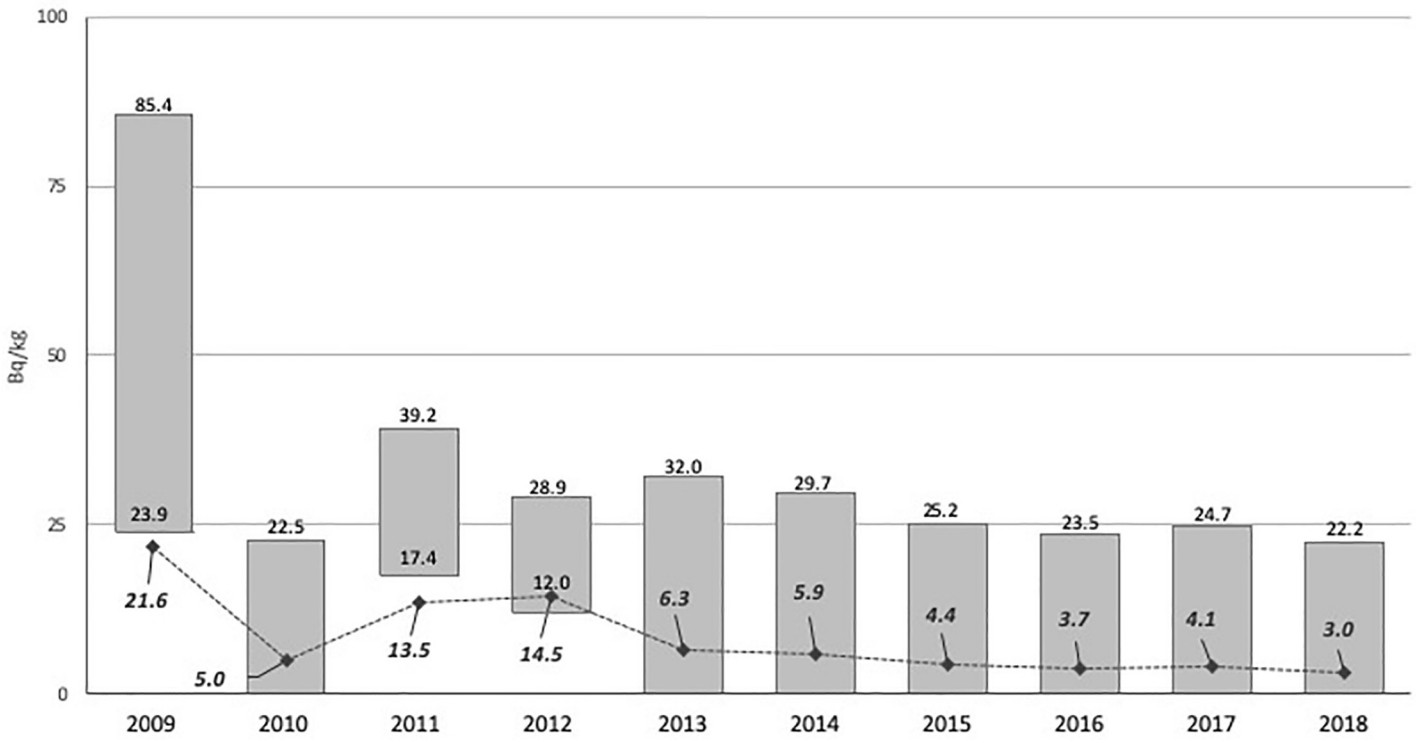

**Fig 1. Mean and the 75th and 95th percentiles of $^{137}$Cs concentration (Bq/kg) in the body.** The top and bottom of each bar indicate the value of $^{137}$Cs concentration in the body at the 75th and 95th percentiles for each year. The stretching line through the bars is the mean. The $^{137}$Cs concentration in the body at the 75th percentile was detectable only in 2009, 2011, and 2012, and thereafter, it had below detectable levels. The body burden decreased gradually with the progress of the study, with small fluctuations.

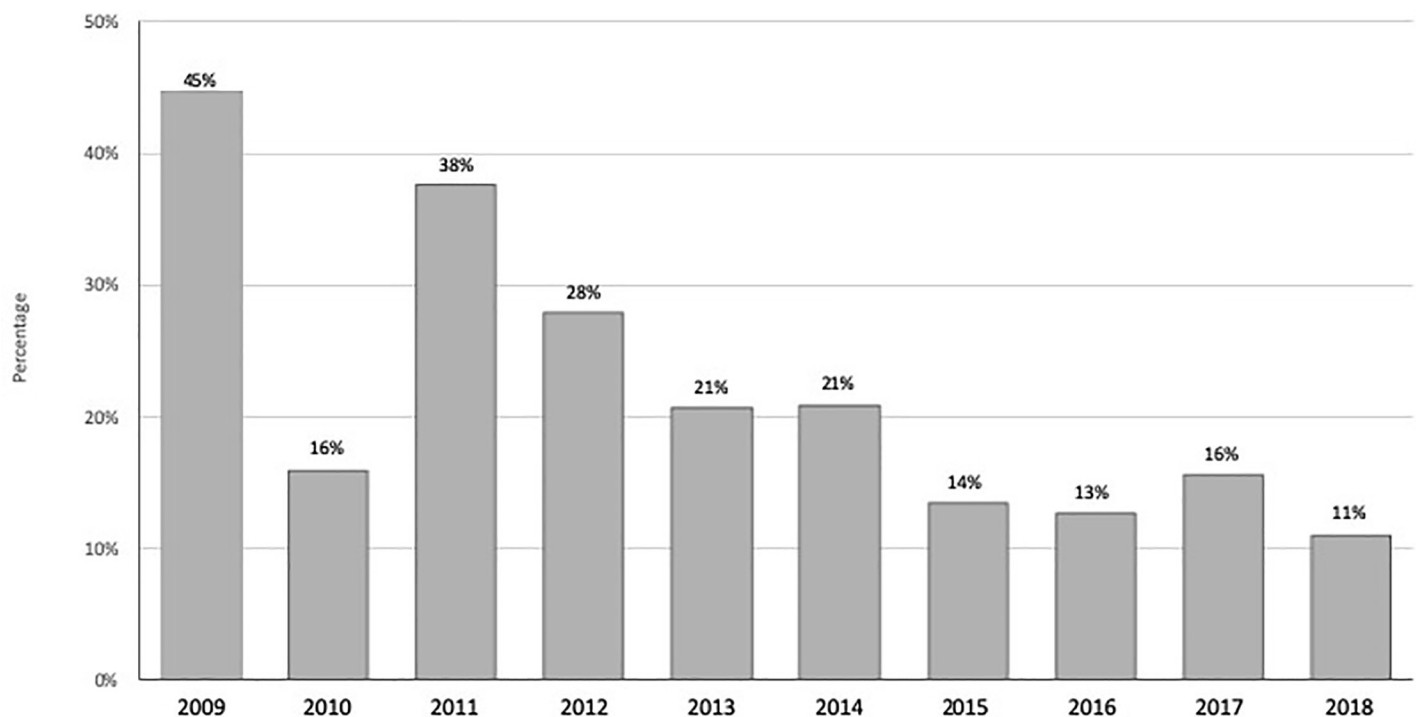

**Fig 2. Frequency of people with detectable levels of $^{137}$Cs from 2009 to 2018.** The frequency of people with detectable levels declined steadily during the study, with moderate fluctuations.

the group of people in each individual year. The results presented in Fig 3 graphically revealed that autumn had the highest number of people with detectable contamination for most years, particularly in the first half of the study, followed by winter. The increased percentage in autumn was significant in 6 years (2009, 2010, 2011, 2012, 2014, 2017) out of 10 years (p<0.05). Winter was the second season with the highest number of people with detectable contamination, though only the increase in three years (2009, 2014, 2018) was significant. Although the seasonal frequency of people detectable with $^{137}$Cs for the whole data set was significant (p<0.001), the chi-square post hoc test performed for each individual year showed that the increase in autumn in two years (2013, 2016) and in winter in seven years (2010, 2011, 2012, 2013, 2015, 2016, 2017) was not significant (p>0.05).

**Table 2. Number and percentage of participants classified by a range of annual effective internal doses (mSv/y).**

|            | 2009   | 2010   | 2011   | 2012   | 2013  | 2014   | 2015   | 2016   | 2017   | 2018   |
|------------|--------|--------|--------|--------|-------|--------|--------|--------|--------|--------|
| < 0.1 mSv  | 7 736  | 11 836 | 10 798 | 10 233 | 9 505 | 10 752 | 10 694 | 11 424 | 12 800 | 10 651 |
|            | 85.8%  | 98.1%  | 95.2%  | 97.7%  | 97.0% | 97.7%  | 98.1%  | 98.8%  | 98.6%  | 98.9%  |
| 0.1 mSv ≤  | 1 248  | 233    | 535    | 240    | 298   | 249    | 204    | 137    | 187    | 123    |
|            | 13.8%  | 1.9%   | 4.7%   | 2.3%   | 3.0%  | 2.3%   | 1.9%   | 1.2%   | 1.4%   | 1.1%   |
| 1 mSv ≤    | 29     | 1      | 2      | 3      | 1     | 4      | 6      | 0      | 0      | 0      |
|            | 0.3%   | 0.0%   | 0.0%   | 0.0%   | 0.0%  | 0.0%   | 0.1%   | 0.0%   | 0.0%   | 0.0%   |
| 5 mSv ≤    | 0      | 0      | 1      | 0      | 0     | 0      | 0      | 0      | 0      | 0      |
|            | 0.0%   | 0.0%   | 0.0%   | 0.0%   | 0.0%  | 0.0%   | 0.0%   | 0.0%   | 0.0%   | 0.0%   |
| 10 mSv ≤   | 1      | 1      | 1      | 2      | 0     | 0      | 0      | 1      | 0      | 0      |
|            | 0.0%   | 0.0%   | 0.0%   | 0.0%   | 0.0%  | 0.0%   | 0.0%   | 0.0%   | 0.0%   | 0.0%   |

**Table 3. Seasonal differences in $^{137}$Cs concentration (Bq/kg) in residents for the whole period and each year separately.**

|  | All | 2009 | 2010 | 2011 | 2012 | 2013 | 2014 | 2015 | 2016 | 2017 | 2018 |
|---|---|---|---|---|---|---|---|---|---|---|---|
| Spring[a] | 6.31 | 21.0 | 3.4[c] | 10.0 | 4.9 | 8.6[b,c,d] | 4.2[b,c] | 5.8[c] | 2.7 | 3.6[b,c] | 3.1[c,d] |
| Summer[b] | 6.30 | 22.3 | 5.9 | 9.2[c] | 5.6 | 4.1[a,c,d] | 7.9[a,d] | 4.4 | 3.3 | 2.5[a,c,d] | 2.8[c,d] |
| Autumn[c] | **10.01** | 22.5 | 6.7[a,d] | 12.0[b] | 40.2 | 5.9[a,b] | 7.1[a,d] | 2.7[a,d] | 3.4 | 6.2[a,b,d] | 1.1[a,b,d] |
| Winter[d] | **8.74** | 20.7 | 4.0[c] | 23.3 | 12.8 | 6.4[a,b] | 4.2[b,c] | 4.5[c] | 5.1 | 3.9[b,c] | 4.9[a,b,c] |

[a,b,c,d] indicates the significance of the difference between the corresponding (indexed) seasons.

Table 4 represents the general data regarding internal exposure among men and women in terms of proportion and frequency of participants with detectable contamination and the mean Bq/kg concentration for the whole data set as well as for the detectable groups only. Although the percentage of women in the entire data set was the highest (70.8%), the proportion of women with detectable contamination in the women's group accounted for 19.4%. As opposed to women, men were a relatively small percentage in the entire data set, though their proportion with detectable contamination was significantly higher (p<0.001), at a rate of 26.8%. Furthermore, the men's group exceeded the women's group in terms of the average value of Bq/kg for the entire data set as well as for the detectable group only (p<0.001).

We also examined the average internal body burden and percentage of people with detectable contamination in different age groups, as presented in Table 5, by dividing the

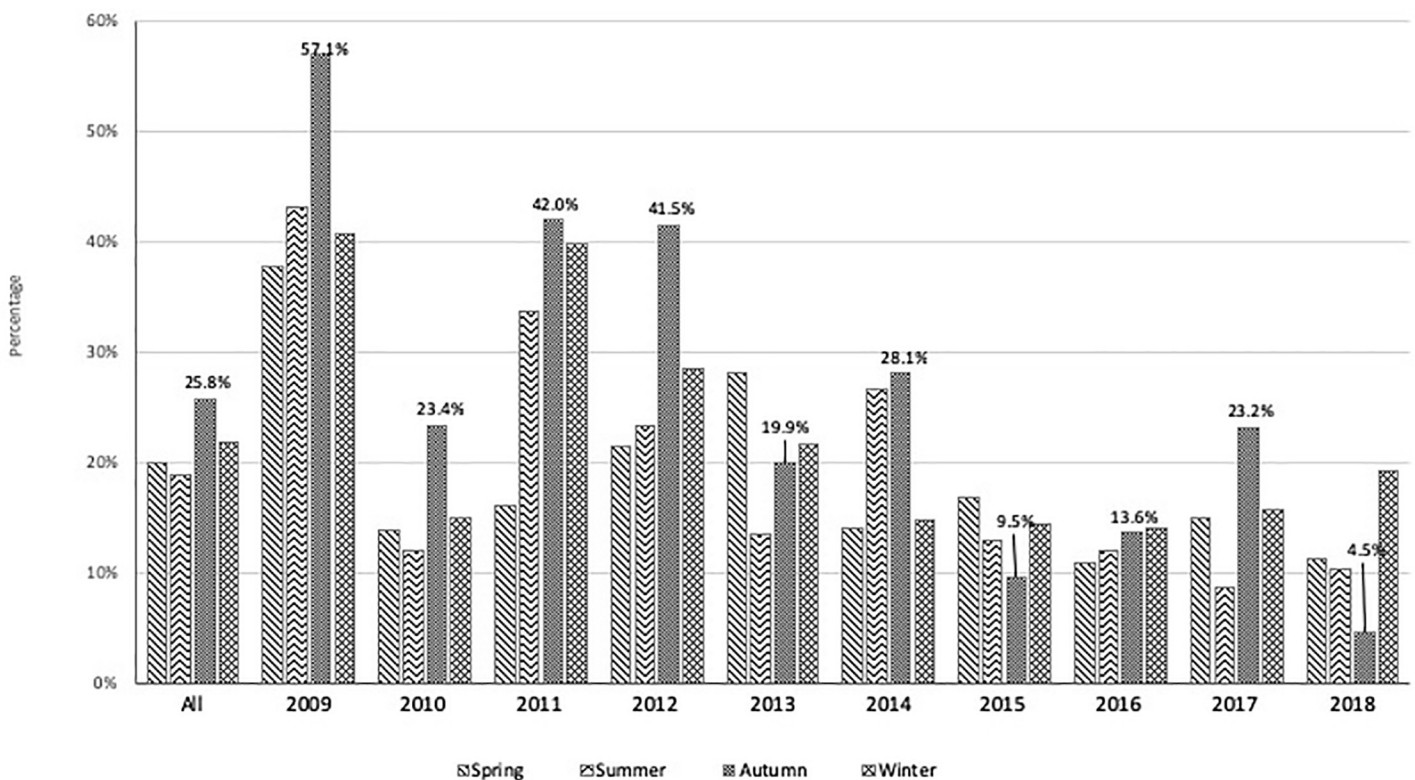

**Fig 3. Seasonal frequency of residents with detectable levels for all data set and each year.** The increased percentage in autumn was significant in 6 out of 10 years (p<0.05). Winter had the second highest rate of frequency of people with detectable contamination, though only the increase in three years (2009, 2014, 2018) was significant.

**Table 4. Population, participants with detectable contamination, and mean $^{137}$Cs concentration (Bq/kg) by sex.**

|  | All | Women | Men |
|---|---|---|---|
| All participants (%) | 109,936 | 77,848 (70.8%) | 32,088 (29.2%) |
| Detectable participants (%) | 23,697 (21.6) | 15,100 (19.4%) | 8,597 (26.8%) |
| Mean Bq/kg in all | 7.82 | 6.6 | 10.7 |
| Mean Bq/kg in detectable group | 36.3 | 34.2 | 40 |

participants into four different age groups: 0–10, 11–20, 21–60, and > 60 years. Statistical analysis revealed that the average $^{137}$Cs concentration (Bq/kg) in the 11–20-year age group was significantly higher than that in the 0–10-year and 21–60-year age groups (p<0.001). The percentage of residents with detectable contamination in each corresponding age group was significantly different (p<0.001), except for those aged 0–10 years.

We also performed the Tamhane test to determine the significance of differences in Bq/kg means in each age group by year, as shown in Table 6. A significant difference was found in the 11–20-year age group in 2016 and 2017 (p<0.05) and in the 0–10-year age group in 2014 and 2016 compared with that in the other age groups (p<0.05).

## Discussion

In this study, the annual average body burden decreased significantly (around 86%) over the duration of the study, from 22 Bq/kg in 2009 to 3 Bq/kg in 2018. The general trend of the frequency of people with detectable levels followed the same pattern, declining from 44.6% to 11.2% during the course of the study. The $^{137}$Cs concentration at the 75$^{th}$ percentile of participants was identified in only the first three years of the study, and it could not be detected from 2012 onwards. Our study revealed that either the average Bq/kg concentration in the bodies of residents of Korosten and its subordinate villages decreased considerably or the number of examination results with detectable levels declined over the course of the study, with fluctuations in some years. The results depicted above are generally consistent with those of previous studies performed in the same and different contaminated territories in the vicinity of the CNPP [4–9].

In the correlation analysis of the associations, we found that the mean $^{137}$Cs concentration in the body had a significantly strong positive correlation with the annual frequency of people with detectable contamination. This finding indicated clearly that the number of people eating contaminated forest food had increased in the years when the mean Bq/kg concentration had increased. The high fluctuations in the body burdens shown in our study may imply that

**Table 5. Population, participants with detectable contamination, mean and standard deviation of $^{137}$Cs concentration (Bq/kg), and women's proportion in each age group.**

|  | All | 0–10[a] | 11–20[b] | 21–60[c] | > 60[d] |
|---|---|---|---|---|---|
| All participants (%) | 109 936 | 3 318 (3.0) | 14 292 (13.0) | 71 543 (65.1) | 20 783 (18.9) |
| Detectable (%) | 23 697 (21.6) | 694 (20.9) | 3 887 (27.2)* | 14 817 (20.7)* | 4 299 (20.7)* |
| Mean (SD) | 7.8 (229) | 5.9[b] (15) | **11.1[a,c] (65)** | 7.0[b] (145) | 8.7 (449) |
| Mean detectable (SD) | 36.3 (491) | 28.4[b] (21) | **40.9[a] (119)** | 33.7 (316) | 42.2 (985) |
| % women in all | 69.7 | 52.4 | **62.2** | 73.5 | 70.5 |

[a,b,c,d]: significance of difference to corresponding group (p<0.05).

* Chi-square Post hoc test.

SD: standard deviation.

**Table 6. Mean $^{137}$Cs concentration (Bq/kg) in each age group from 2009 to 2018.**

| Age | 2009 | 2010 | 2011 | 2012 | 2013 | 2014 | 2015 | 2016 | 2017 | 2018 |
|------|------|------|------|------|------|------|------|------|------|------|
| 0–10[a] | N/A | 4.6 | N/A | 5.3[b] | 7.3 | 8.2[b,c] | 5.7[d] | 7.0[b,c,d] | 4.0 | 2.5 |
| 11–20[b] | 43.5 | 4.7 | 15.2[d] | 11.0[a] | 6.8 | 5.2[a] | 5.8[d] | 4.4[a,d] | 5.3[c,d] | 2.8 |
| 21–60[c] | 17.0 | 5.4 | 14.6 | 8.3 | 5.9 | 5.7[a] | 4.3 | 3.6[a] | 4.0[b] | 3.0 |
| > 60[d] | 15.1 | 3.6 | 9.0[b] | 41.2 | 6.6 | 6.2 | 3.4[a,b] | 2.8[a,b] | 3.6[b] | 3.0 |

[a,b,c,d] Significance of difference to corresponding group (p<0.05).

N/A-not applicable (sample for 0–10 in 2009 is 1 and 3 for 2011).

residents of contaminated areas probably changed their dietary habits and allowed themselves and their family members to eat contaminated food during some years depending on the circumstances in that specific year. This may be related to their attempts to alleviate food expenses and diversify the variety of meals, for example, when they experienced economic problems or when the forest foodstuffs were abundant. Years of plentiful forest food may have tempted most people living in the vicinity of the forest to collect them frequently. Some field researchers have said that villagers usually collected much more forest food than they could consume, with the aim of selling the surplus to the nearest city residents, in a bid to earn revenue.

This 10-year screening study detected only 53 people (0.02%) who had an annual internal effective dose exceeding 1 mSv, the dose limit set by the International Commission on Radiological Protection for the general public [11]. This rate was notably small compared with that in a previous WBC screening study that was conducted (513, 0.35%) in 1996–2008 at the same location [6]. This previous study and our study had almost the same number of subjects and study periods was similar. Twenty-seven out of 53 were villagers, while the others were from Korosten city. Given that the proportion of villagers in the entire data set was only 36.4%, the aforementioned number of villagers with high doses implied that people living in villages were likely to experience higher exposure. The number of people with a dose between 1 mSv and 5 mSv was 29 in 2009, which was relatively higher than in the other years of the study. We compared the number of cases for the aforementioned dose range with the results of several preceding years (2000–2009) from the previous study, and it showed only one to six cases in each individual year [6]. Therefore, we found that the number of people with doses between 1 mSv and 5 mSv increased to a greater extent in 2009 than in the preceding year. The spike in the numbers in 2009, and the restoration of the level returned to the previous level in the following year, indicated that 2009 could have been the year when there was abundant forest foodstuff, or possibly some economic changes.

Despite the results and conclusions of similar studies varying slightly from one another, most $^{137}$Cs dose estimations are in agreement with other similar studies in terms of decreasing magnitudes and their patterns [3–9]. Lower dose estimations are expected every year because of the physical decay and vertical migration of radionuclides in soil matrices over the difference in study periods [3]. Apart from the physical and natural peculiarity of $^{137}$Cs mentioned above, Bernhardson et al. also emphasized that "The changes in the WBC values can be large from year to year for several reasons, such as the amount of available forest food or participants with different attitudes in terms of following the food recommendations. In addition, despite the fact that the group of measured persons was composed of similar inhabitants (in terms of age, sex, region) in each village and year, it was necessary for the same individuals to be measured from year to year. This may partly explain some of the year-to-year variability in the observed WBC in the villages" [9]. A large population with broader coverage may affect the mean and median value or other statistical values due to greater heterogeneity of participants

in terms of soil contamination, diet habits, accessibility to forest, level of income, and wider distribution of samples.

Since our study covered a long time period with a considerably limited population and was based on voluntary intention, it can be assumed that some of the examinees underwent screening several times. The number of examinees and their values of internal exposure could also have been influenced by the socioeconomic situation each year, which affected the purchasing power of their income, since visiting the hospital from remote villages involved financial transport expenses. During the 1990s, the average annual effective dose reduced from 2.5 mSv to 0.3 mSv, whereas the importance of internal exposure had been progressively increasing, as its contribution to the total effective dose started dominating [9]. It has been expected, from a long-term perspective, that the internal dose after the CNPP accident would be highly dependent on the deposition of 137Cs, as people tended to forget about the protective measures, and in addition the economic situation may force them to rely more on locally produced food [4]. In addition, an investigation conducted in Japan after the accident at Fukushima Daichi Nuclear Power Plant (FDNPP) revealed that the mean value of internal radioactivity in residents has been steadily increasing at a slow pace from 2012 to 2015, due to the elderly people's willingness to enjoy forest food despite their awareness its contamination [12].

In the seasonal analysis, the mean Bq/kg for each season of the whole data set was 6.31, 6.30, 10.1, and 8.74 for spring, summer, autumn, and winter, respectively. Even though the mean values of autumn and winter were higher, the differences between the seasons were not statistically significant (p>0.05). Yearly post hoc analysis showed significant differences in seasons only for a few years. However, Fig 3, which represents the frequency of people with detectable contamination in each season, demonstrated a higher percentage in winter and autumn for most years of study. The entire data set accounted for 19.9%, 18.8%, 25.8%, and 21.9% for spring, summer, autumn, and winter, respectively with a significant difference between seasons. Post hoc tests for each year showed a significant difference in percentages between autumns of six years and winters of three years compared to other seasons. The seasonal effect is another domain that could raise a question about consistency with other studies and remains under the focus of most dedicated scientists in this field. However, detecting the seasonal effect becomes more challenging as the dietary patterns of the people depends on several factors, and is subject to change. In addition to the continuous physical decay of cesium and fluctuation of socioeconomic conditions, years with abundant wild food have made the observation of seasonal effects complicated. Even though we were able to see a significantly higher mean and proportion of examinees with detectable levels in autumn and winter for all data sets, it was not fully explicit for each year. The main studies reporting investigations of seasonal effects that have been conducted in recent years encompassing larger populations [3, 5, 6], reported insignificant seasonal effects. However, early phase studies with relatively fewer populations, which focused on highly contaminated areas, in contrast, reported a significantly strong existence of the seasonal effect, mainly pointing to autumn and sometimes to winter [7, 8, 13].

Seasonal effects are directly associated with the consumption of forest wild food, such as berries, mushrooms, wild animal meat, and fish, which usually contain high amounts of 137Cs. The products listed above contain much higher amounts of 137Cs than kitchen-garden products [14–17]. Studies evaluating a diet of contaminated food reported that natural products from contaminated forests contributed effectively to the increase of 137Cs concentration in the body during the harvest period, even 30 years after the accident. Similar to a few other previous papers, we could not find strong evidence of the seasonal difference in mean Bq/kg; however, there was a significantly high proportion of people with detectable levels in autumn.

In the body burden analysis with respect to sex, men exceeded women in all parameters except the number of participants, which comprised only 29.2%. The mean internal exposure

in the men's group and only those with detectable contamination were 10.7 and 40 Bq/kg, respectively, while the corresponding values in the women's group were relatively lower at the level of 6.6 and 34.2 Bq/kg. The number of detected participants in the men group was expectedly higher and accounted for 26.8%, outreaching the women group to a great extent, which was only 19.4%. The difference in concentration level of $^{137}$Cs between sex was nearly similar to most previous studies, showing that men have significantly higher internal exposure as well as a higher number of detected. The average $^{137}$Cs concentration in men exceeded that in women by a factor of 1.6, which is slightly lower than 1.7 and 2 found in Bryansk [14] and Norway [18], respectively. Because $^{137}$Cs accumulates in the muscles and bones, its concentration is related to the size of the body. This characteristic was relevant even when the annual internal dose decreased gradually. The reasons for this difference include the slower excretion of $^{137}$Cs from men's bodies and the potentially larger quantities of consumption of mushrooms and other foods, such as wild animals and fish [19].

Additionally, we investigated the mean Bq/kg values in children, adolescents, adults, and elderly people for all data set as well as for each individual year, which revealed higher mean concentrations in children and adolescents than in adults and older adults, with less standard deviation (SD), which is contrary to previous studies. The adolescent group had a significantly higher internal concentration than the other groups, presumably because adolescents consume more food considering kilogram food consumed to body weight. Another possible reason for this finding is that the proportion of men in the groups of children and adolescents was larger than that in other age groups, which may have also led to higher levels. Children and adolescents generally consume a larger quantity of food per kilogram of body mass than adults, reflecting their higher metabolism. Therefore, children may have a larger Bq intake. They may also have a slightly different diet with specific consumption of potentially more contaminated products (e.g., milk). However, Bernhardsson et al. examined body burden in the Bryansk region from 2006 to 2008 and reported that the lowest amount of $^{137}$Cs was in children, although the difference between adults and children was not large [9]. The same has been reported in an early study by Thronberg et al. implemented during 1991–2000, that is, that the average body concentration was significantly lower in children, assuming that adults consume larger amounts of mushrooms than children [4]. Low internal exposure levels in children have been explained by rapid biological removal, that is, metabolism of internal $^{137}$Cs [20], and is attributed partly to two supplemented uncontaminated meals a day provided to the children in contaminated territories during the school year by the local governments [4].

In contrast to these previous study results, children and adolescents had higher internal concentration levels in our study. This disagreement may have arisen because of the number of samples in each age group and the sex ratio. The number of participants in each age group in our study were 3 318, 14 292, 71 543, and 20 783, for children, adolescents, adults, and elders, respectively. Therefore, a large number of adults and fewer participants in other groups may have influenced the average values of body burden. The same effect might have occurred because of sex ratio, which varied significantly, with a larger proportion of men in the children and adolescents' groups. Therefore, for further understanding, we calculated the proportion of people with detectable contamination in each corresponding age group (children 20.9%, adolescents 27.2%, adults 20.7%, and older adults 20.7%) and found again that the adolescents group had the highest significant percentage. This provided additional evidence that adolescents had considerably higher body burden and a greater proportion of participants with detectable contamination. We also examined the mean concentration levels among those who had detectable concentrations and obtained slightly different results, that is, the children's group showed the lowest mean Bq/kg and the adolescents group had the second highest mean Bq/kg after the elderly group, although there were no significant differences between the

groups. The finding of high levels of radioactivity among the older adults group was partly consistent with a recent study by Japanese scientists that was conducted in contaminated areas affected by fallouts from the FDNPP after the accident, which found that the majority (83.2%) of individuals with detectable radioactivity were seniors, over 50 years old, with men accounting for 68%, suggesting that the elderly men group prefers to enjoy the rest of their lives eating diverse wild plants, while fully recognizing the threat of internal radiation [12].

This study has several limitations. First, duplication of measurements could have taken place, since this study assessed the body burden of $^{137}$Cs in residents on a voluntary basis who requested measurement; none were excluded from the analysis. However, taking into account the biological half-life of $^{137}$Cs, we assumed that the measurement indications of each participant in each year could be counted per participant. Second, some other possible bias may have existed in the selection of study participants, as the subjects were those who came to the center voluntarily and were examined. Presumably, the nonuniform dissemination of information within the research area, financial hardships, and living in the farthest villages may have led to a nonuniform geographic distribution of participants. Considering women's risk attitude toward health care may lead to another bias [21]. The sex ratio in our study, where around 71% accounted for women, clearly showed that women tended to be more concerned about health conditions, and therefore visited hospitals more frequently. Third, the study did not follow each person's internal dose; neither interviews nor questionnaires were involved in identifying the lifestyle, dietary habits, or period of residence. We also did not observe and validate the level of food contamination and decontamination measures in the study settings.

In conclusion, in comparison to previous studies, very few people had excessive doses during the entire study period, and almost all of the measured population were below the regulatory dose limit. Despite this, the frequency of the population with detectable levels remained at a considerably high level. The frequency of people with detectable levels and their fluctuations indicated that the residents were still dependent on locally produced food and forest products, which may be more critical in years of economic recession. Thus, we will have to continue with our efforts to monitor recent changes in contaminated areas with regard to the body burden, and consequently share our findings with the public, experts, and related specialists to prevent, protect, and predict future circumstances. This will also encourage residents to be more prudent and aware of internal radiation levels, sources, and pathways. The awareness of internal exposure in contaminated areas will encourage residents to moderate their intake of foodstuffs from the forests and also consequently urge them to measure their internal radiation exposure in a systematic way, even though the level of contamination in the affected areas is gradually decreasing. The observed high fluctuations in internal exposure and frequency during some years led us to assume that there may have been years of plentiful forest food or substantial adverse economic impact, which prompted residents to consume more local and forest products. The relatively high fluctuations recorded during this study need to be studied more thoroughly to understand the main reasons and occasions, which in turn would improve preventive measures and contribute to better management of similar circumstances and developments in the future. More comprehensive investigations are necessary to identify the details of the pathways of the internal body burden, which may be essential in providing persuasive information for risk communication and relief of anxiety among residents living in contaminated zones.

## Acknowledgments

The authors would like to thank the staff at the Zhytomyr Inter-Area Medical Diagnostic Center and all the personnel who participated in this study. We would also like to thank Dr. Maika

Nakao for her guidance. We would like to thank Editage (www.editage.com) for English language editing.

## Author Contributions

**Conceptualization:** Yesbol Sartayev, Jumpei Takahashi, Alexander Gutevich.

**Data curation:** Yesbol Sartayev, Jumpei Takahashi.

**Formal analysis:** Yesbol Sartayev.

**Funding acquisition:** Yesbol Sartayev, Alexander Gutevich.

**Investigation:** Yesbol Sartayev, Naomi Hayashida.

**Methodology:** Yesbol Sartayev, Jumpei Takahashi, Alexander Gutevich, Naomi Hayashida.

**Project administration:** Naomi Hayashida.

**Resources:** Alexander Gutevich.

**Software:** Yesbol Sartayev, Naomi Hayashida.

**Supervision:** Naomi Hayashida.

**Validation:** Naomi Hayashida.

**Writing – original draft:** Yesbol Sartayev.

**Writing – review & editing:** Yesbol Sartayev, Jumpei Takahashi, Alexander Gutevich, Naomi Hayashida.

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
