## [Decision Letter · Decision Letter 0]

16 Nov 2020

PONE-D-20-16627

Screening for 137Cs body burden due to the Chernobyl accident in Korosten city, Zhytomyr, Ukraine: 2009-2018

PLOS ONE

Dear Dr. Hayashida:

Thank you for submitting your manuscript to PLOS ONE. After careful consideration, we feel that it has merit but does not fully meet PLOS ONE’s publication criteria as it currently stands. Therefore, we invite you to submit a revised version of the manuscript that addresses the points raised during the review process.

Minor revisions were recommended by both reviewers. Please see comments below.

We look forward to receiving your revised manuscript.

Kind regards,

Gayle E. Woloschak, PhD

Academic Editor

PLOS ONE

Journal Requirements:

3. We note that Figure 1 in your submission contains map images which may be copyrighted.

We require you to either (a) present written permission from the copyright holder to publish this figure specifically under the CC BY 4.0 license, or (b) remove the figure from your submission:

b. If you are unable to obtain permission from the original copyright holder to publish this figure under the CC BY 4.0 license or if the copyright holder’s requirements are incompatible with the CC BY 4.0 license, please either i) remove the figure or ii) supply a replacement figure that complies with the CC BY 4.0 license. Please check copyright information on all replacement figures and update the figure caption with source information. If applicable, please specify in the figure caption text when a figure is similar but not identical to the original image and is therefore for illustrative purposes only.

4. Please upload a new copy of Figure 1 as the detail is not clear. Please follow the link for more information: https://blogs.plos.org/plos/2019/06/looking-good-tips-for-creating-your-plos-figures-graphics/

Additional Editor Comments:

Minor changes were recommended by both reviewers.

Reviewers' comments:

Reviewer's Responses to Questions

**Comments to the Author**

1. Is the manuscript technically sound, and do the data support the conclusions?

Reviewer #1: Yes

Reviewer #2: Yes

2. Has the statistical analysis been performed appropriately and rigorously? 

Reviewer #1: Yes

Reviewer #2: I Don't Know

3. Have the authors made all data underlying the findings in their manuscript fully available?

Reviewer #1: Yes

Reviewer #2: No

4. Is the manuscript presented in an intelligible fashion and written in standard English?

Reviewer #1: Yes

Reviewer #2: No

5. Review Comments to the Author

Reviewer #1: The manuscript is an excellent research finding and statistical analysis and discussion is also excellent. The research finding of the manuscript will be huge attraction to the professional who are working in the field of radiation protection and radiation safety.

Author need to take into consideration the following:

(1) In abstract line no. 22, replace radioactive materials by fission products;

(2) Reference should be revised as per PLOS ONE journal style;

Reviewer #2: This paper presents new and valuable information concerning cesium body burdens in the population near the Chernobyl accident. However, the English translation is flawed and considerable editing is required. In particular, statements regarding statistical significance are not always comprehensible.

Line 69-70: "...received annual effective doses of less than 1 mSv in 137Cs addition to the natural background doses of

from the Chernobyl fallout." This reads as if Chernobyl fallout were natural background.

Line 74-75: "... reported a high frequency of people with an estimated internal effective dose.." This required editing; zero can be an estimated dose.

Line 77: "...the effective dose of internal exposure is becoming increasingly important..." This phraseology is misleading. The fraction of the dose from internal exposure is becoming larger, but it is a larger fraction of a decreasing dose.

Line 104-106: The concept of "contamination level zones" required some introduction. Simply referring to smaller digits is not clear. Add a sentence or two to describe the classification of "levels of contamination".

Line 116-119: It is not clear if this is a font change or if the small print is a part of the caption for Figure 1. The concepts of "contamination levels" is not clear (and are not illustrated in Figure 1.)

Line 124: Does the participant lean his abdomen on the seat?

Line 140: This "dose coefficient" requires a reference. Does it assume uniform distribution of Cs in the body? The term "dose coefficient" in the health physics literature has a different meaning (usually Sv per Bq intake). What are the underlying assumptions? - continuous intake and contamination at that level? For what period - one year?

Line 148 et seq.: How were means calculated? A very large fraction of measurements are below detection (the median is always less than detectable); were they assigned zero ? A minimum level? A log-extrapolated mean?

Line 177-178 and similar places below: "The 75th percentile was detected only for 3 years in the earlier part of the study," Awkward wording. Perhaps "More than 25 percent of measurements exceeded the detection level only for 3 years..."

Line 185-186: Same comment as above. "More than 25% of the measurements exceeded the minimum detectable level only in 2009, 2011, and 2012..."

Line 206-207: "all belonging to various areas." Awkward wording. Perhaps "..from various locations in the study domain."

Line 228 and often thereafter: "detectable people". Change to "people with detectable contamination"

Also line 236, 240, 243, 244, Table 4, 251-251, 257, 262, 389, 390, 430, 433-434

Line 231: Discussions of statistical significance in this paper are difficult to follow - I suspect that the translator was not familiar with the concept. Here, what does "were insignificant" refer to? Similarly Line 272, 358

Line 254: "...average internal Bq/kg body burden..." This is poor wording. Perhaps "average Cs tissue concentrations (Bq/kg)..." or something like that.

Line 285-286: The 75th percentile for Bq/kg was identified in only 3 different years at the beginning part of the study." Again, there was always a 75th percentile - it was at a level below detection. Awkward wording.

Line 295: "...and showed significance." Perhaps "...with statistical significance."

Line 308-309: ICRP does not set regulatory limits, only generic recommendations. The UA government sets the applicable regulations... what are they?

Line 362-363: What does significance imply here?

Line 367-368: "... cesium and socio-economic conditions..." this reads as if socioeconomic conditions were continuously decaying...

Line 379-380: forest products are also "locally produced" (perhaps more so than store-bought foods!)

Line 394-395: "The average Cs137 concentration in men exceeded THAT IN WOMEN by a factor of 1.6"

Line 399: "The reasons for this difference include slower excretion of Cs137 from the male body and POTENTIALLY larger quantities of consumption..." You are making an assumption here about the reason - you did not measure it or ask dietary questionnaires. Similar comment in other nearby locations where you are speculating on the reasons.

Line 441: individuals do not "exhibit radioactivity"

Line 447: "...which is much shorter..." Perhaps "which is short in relation to the counting intervals"

Line 451: "ununiform" should be "nonuniform". Also on line 453

6. PLOS authors have the option to publish the peer review history of their article (what does this mean?). If published, this will include your full peer review and any attached files.

Reviewer #1: **Yes: **Dr. Mohammad Sohelur Rahman, Chief Scientific Officer, Health Physics Division, Atomic Energy Centre Dhaka, Bangladesh, e-mail: msrahman74@hotmail.com;msrahman74@gmail.com; cell: +880-1827328238

Reviewer #2: No

---

## [Author Response · Author response to Decision Letter 0]

20 Dec 2020

Dear Reviewers,

Thank you for inviting us to submit a revised draft of our manuscript.

We also appreciate the time and effort you have dedicated to providing insightful feedback on ways to strengthen our paper. Thus, it is with great pleasure that we resubmit our article for further consideration. We have incorporated the changes that reflect the detailed suggestions you have so graciously provided. We also hope that our edits and the responses we provide in 'Response to Reviewer' file satisfactorily address all the issues and concerns you have noted.

---

## [Decision Letter · Decision Letter 1]

2 Jan 2021

Screening for the 137Cs body burden owing to the Chernobyl accident in Zhytomyr region, Ukraine: 2009–2018

PONE-D-20-16627R1

Dear Dr. Hayashida:

We’re pleased to inform you that your manuscript has been judged scientifically suitable for publication and will be formally accepted for publication once it meets all outstanding technical requirements.

Kind regards,

Gayle E. Woloschak, PhD

Section Editor

PLOS ONE

Additional Editor Comments (optional):

Reviewers' comments:

Reviewer's Responses to Questions

**Comments to the Author**

1. If the authors have adequately addressed your comments raised in a previous round of review and you feel that this manuscript is now acceptable for publication, you may indicate that here to bypass the “Comments to the Author” section, enter your conflict of interest statement in the “Confidential to Editor” section, and submit your "Accept" recommendation.

Reviewer #1: All comments have been addressed

Reviewer #2: All comments have been addressed

2. Is the manuscript technically sound, and do the data support the conclusions?

Reviewer #1: Yes

Reviewer #2: Yes

3. Has the statistical analysis been performed appropriately and rigorously? 

Reviewer #1: Yes

Reviewer #2: Yes

4. Have the authors made all data underlying the findings in their manuscript fully available?

Reviewer #1: Yes

Reviewer #2: No

5. Is the manuscript presented in an intelligible fashion and written in standard English?

Reviewer #1: Yes

Reviewer #2: Yes

6. Review Comments to the Author

Reviewer #1: (No Response)

Reviewer #2: I have no further comments; the responses adequately address my prior comments.

(This form requires >100 characters, so I am still typing!)

7. PLOS authors have the option to publish the peer review history of their article (what does this mean?). If published, this will include your full peer review and any attached files.

Reviewer #1: **Yes: **Dr. Mohammad Sohelur Rahman, Chief Scientific Officer, Health Physics Division, Atomic Energy Centre, Shahbag, Dhaka-1000, Bangladesh, e-mail: msrahman74@gmail.com

Reviewer #2: **Yes: **Bruce A. Napier

---

## [Editor Report · Acceptance letter]

7 Jan 2021

PONE-D-20-16627R1 

Screening for the ^137^Cs body burden owing to the Chernobyl accident in Zhytomyr region, Ukraine: 2009–2018 

Dear Dr. Hayashida:

I'm pleased to inform you that your manuscript has been deemed suitable for publication in PLOS ONE. Congratulations! Your manuscript is now with our production department. 

Kind regards, 

on behalf of

Dr. Gayle E. Woloschak 

Section Editor

PLOS ONE